# Are most published research findings false in a continuous universe?

**Kleber Neves** \*, **Pedro B. Tan, Olavo B. Amaral**

Institute of Medical Biochemistry Leopoldo de Meis, Federal University of Rio de Janeiro, Rio de Janeiro, Brazil

\* kleber.neves@bioqmed.ufrj.br

**Data Availability Statement:** The model was developed in R 3.6.3 (R Core Team, 2020). Code for the model, along with auxiliary scripts used to make graphs and analyses, are available on GitHub (https://github.com/KleberNeves/reproducibility-

## Abstract

Diagnostic screening models for the interpretation of null hypothesis significance test (NHST) results have been influential in highlighting the effect of selective publication on the reproducibility of the published literature, leading to John Ioannidis' much-cited claim that most published research findings are false. These models, however, are typically based on the assumption that hypotheses are dichotomously true or false, without considering that effect sizes for different hypotheses are not the same. To address this limitation, we develop a simulation model that overcomes this by modeling effect sizes explicitly using different continuous distributions, while retaining other aspects of previous models such as publication bias and the pursuit of statistical significance. Our results show that the combination of selective publication, bias, low statistical power and unlikely hypotheses consistently leads to high proportions of false positives, irrespective of the effect size distribution assumed. Using continuous effect sizes also allows us to evaluate the degree of effect size overestimation and prevalence of estimates with the wrong sign in the literature, showing that the same factors that drive false-positive results also lead to errors in estimating effect size direction and magnitude. Nevertheless, the relative influence of these factors on different metrics varies depending on the distribution assumed for effect sizes. The model is made available as an R ShinyApp interface, allowing one to explore features of the literature in various scenarios.

## Introduction

In 2005, John Ioannidis published an article with the alarming conclusion that most published findings in biomedical science are false [1]. The work, which has received thousands of citations since then, was a turning point for the awareness of reproducibility issues in science, which has steadily increased over the last two decades in a process that has been viewed both as a "reproducibility crisis" and as a "credibility revolution" [2, 3].

The paper's conclusion is based on an analytical model of the scientific discovery process, largely inspired by the framework used in the interpretation of clinical diagnostic tests (for historical references, see [4]). In this framework, hypotheses are either true or false, and scientists testing them find positive or negative results on the basis of null-hypothesis significance tests.

model). The generated data for the simulations
shown are available at https://osf.io/z2hn9/files/.
The model is available as a Shiny App at https://
kneves.shinyapps.io/repro-model/.

**Funding:** This work was supported through grants
from FAPERJ (E-26/203.222/2017; OBA) and the
Serrapilheira Institute (OBA; KN. PBT). The funders
had no role in study design, data collection and
analysis, decision to publish, or preparation of the
manuscript.

**Competing interests:** The authors have declared
that no competing interests exist.

The results will thus depend both on the proportion of true to false hypotheses and on the
error rates of the tests, which are combined using Bayes' theorem. Assuming values for the
model's parameters in different scenarios of biomedical research, Ioannidis calculates that the
positive predictive value (PPV; the proportion of true positives among positive results; see
Methods for details) of published medical research findings would be below 50% in most
cases.

Despite its broad influence, the study has also attracted a fair amount of criticism. Critics of
the model have pointed out its pessimistic assumptions about the prevalence of bias [5] and
the low proportion of true hypotheses assumed for some scenarios [6, 7]. After all, even with
well-powered and unbiased research, one can still get a very low PPV with a low enough preva-
lence–with the converse also being true, as long as the prevalence is high enough [4]. The
model also makes implicit–and pessimistic–assumptions about publication bias and interpre-
tation of individual findings in isolation that do not necessarily hold for all fields of science. By
exploring the effects of removing some of these assumptions, other authors have found a less
dreary picture of the literature [8–10].

A more basic concern, however, is that the model makes unrealistic assumptions about the
nature of scientific hypotheses, which are treated as dichotomously true or false [4]. In a real-
life scenario, however, it makes no sense to consider statistical power as constant for all true
effects, which essentially assumes that they all have the same size, or consider that all false
hypotheses have effect sizes strictly equal to zero. On the contrary, effect sizes in most fields
are likely to follow a continuous distribution, [11–13], and while NHSTs are designed to detect
scientifically relevant effects, it is not the case that all relevant effects are of similar magnitude,
or that all other effects are null [14]. As Wilson and colleagues [13] put it, the framework is at
most a useful fiction that can mislead if misused.

To address these criticisms and investigate whether the conclusions of Ioannidis [1] hold
true in more realistic scenarios, we developed a simulation model that expands upon their
original premises. Our model assumes continuous effect sizes following different distributions,
which are classified as true or false based on minimum effect sizes of interest. Besides evaluat-
ing positive predictive values, we also analyze p-value distributions and other measures of
accuracy, such as sign and magnitude errors [14], under different scenarios of prevalence, sta-
tistical power and bias. The model is available as an R ShinyApp interface that allows the user
to explore features of the literature under various assumptions about the scientific discovery
process.

## Results

The different effect size distributions used in the model are shown in Fig 1. We initially use a
dichotomous distribution of hypotheses as in Ioannidis [1], with explicit effect sizes for "true"
and "false" hypothesis (0 and 1, respectively; Dichotomous model, Fig 1A). The process of sim-
ulating experiments is summarized on S1 Fig. In each experiment, scientists perform t-tests
between two experimental groups. An effect size of 0 or 1 is selected randomly for each experi-
ment with a probability based on the prevalence of true hypotheses. Samples are then gener-
ated from normal distributions centered at zero (control) or at the effect size (treatment), both
with SD = 1 to reflect sampling error, with a sample size calculated to yield a given statistical
power to detect a typical true effect. The difference between group means is used as the effect
size estimate for each experiment, and results are published if significant results (p<0.05) are
found. When the initial results are non-significant, the experiment has a probability of being
repeated according to a bias parameter until a significant result is obtained and published.
Although we model bias by repeating experiments, it can be thought of as encompassing other

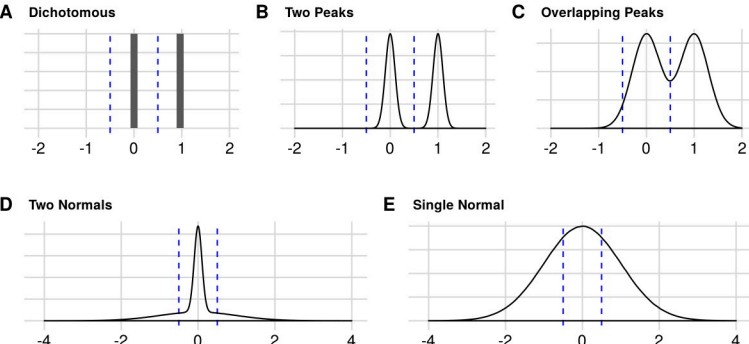

**Fig 1. Effect size distributions used in the simulations.** In the Dichotomous model (**A**), the density curve is replaced by a histogram, as effects sizes are strictly 0 or 1. The Two Peaks (**B**), Overlapping Peaks (**C**) and Two Normals (**D**) models are built as a mixture of two normal distributions with different means and/or SDs (see Methods for details). The weight parameter that regulates the mixture is 0.5 in the examples shown, leading to equal weights for both curves–in simulations, however, it varies according to the prevalence of true effects. For the Single Normal model (**E**), the weight parameter equals 1 and the standard deviation is used to control the prevalence. Dashed lines indicate the minimum effect size of interest (held fixed at 0.5 for all simulations): effects are considered true if their absolute size is larger than this threshold.

ways to find significance in initially non-significant results, such as p-hacking using multiple analysis options [15]. Simulations are performed until the literature contains 5,000 results.

The positive predictive value (PPV) of published findings is calculated for this dichotomous distribution under different combinations of statistical power, bias and prevalence of true effects (Fig 2A). Increasing statistical power leads to higher PPVs, as more true effects are detected by NHSTs. Increasing bias, on the other hand, markedly lowers the PPV, especially at lower prevalences, as more false positives are generated when bias is prevalent. Our simulation results are in line with those obtained analytically by Ioannidis [1]–for a more direct comparison, the same curves expressing the likelihood of true hypotheses in odds (as in the original publication) are shown in S2 Fig.

A dichotomous distribution containing only null and large effects makes it trivial to define true and false hypotheses; in reality, however, a real scientific field is likely to have an unknown distribution of continuous effect sizes. To model this, we use different mixtures of normal distributions, varying their spreads, centers and relative contributions to the population of effects. To distinguish true from false effects in this scenario, one must introduce a minimum effect size of interest, above which hypotheses are considered true. This parameter can be interpreted as the effect size that would be considered relevant to support a given hypothesis, and is set at 0.5 for all simulations. Sample size calculations in this case are performed to detect an average true effect, defined as the mean of all true effects (i.e. those above the minimum of interest) within the distribution. That said, in accordance with typical practices in most scientific fields (at least within the biomedical sciences), statistically significant results are published as positive even if their estimate is below the minimum effect size of interest (although this only happens when statistical power is high).

We first expand our initial model to a Two Peaks model (Fig 1B), which maintains the dichotomy between null and true effects, but includes some variation in effect sizes. In this scenario, effect sizes are drawn from Gaussians centered at 0 or 1 with an SD of 0.1. The relative weight of each curve is used to vary the prevalence of true effects. This was done to introduce continuous effect sizes in the model, while still closely resembling the dichotomous scenario in terms of clearly distinguishable "true" and "false" effects. As expected, PPV curves for two

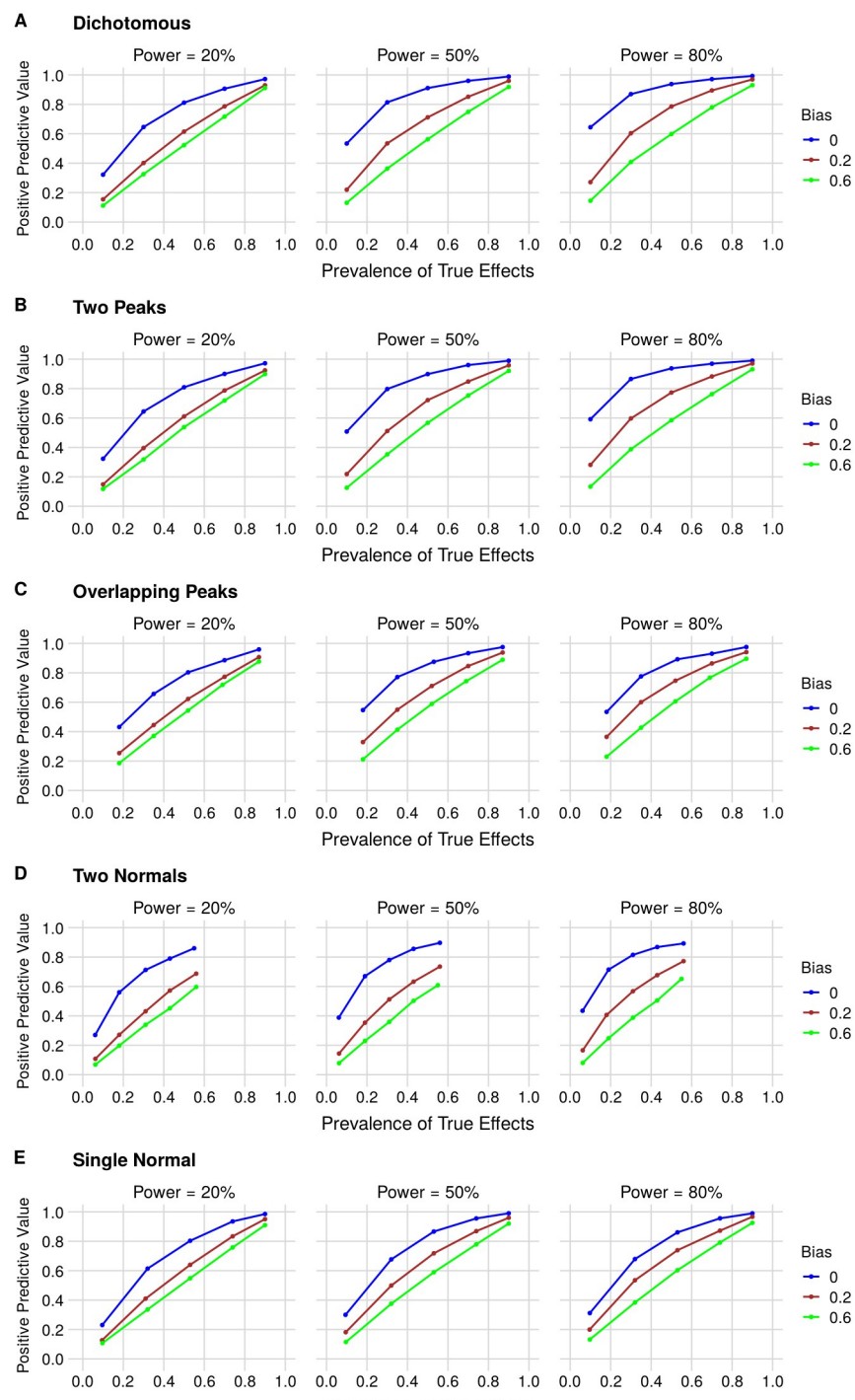

**Fig 2. PPV as a function of the prevalence of true effects, statistical power and bias for (A) Dichotomous, (B) Two Peaks, (C) Overlapping Peaks, (D) Two Normals and (E) Single Normal effect size distributions.** Each point corresponds to a simulation with 5,000 published findings. True effects are defined as those above the minimum of interest (Cohen's $d > 0.5$ in all simulations). In each panel, graphs correspond to 20%, 50% and 80% power from left to right. Blue, red and green curves correspond to bias (i.e. prevalence of negative results that are repeated until becoming positive and then published) of 0, 0.2 and 0.6, respectively. Prevalence ranges vary among scenarios, as for some distributions some prevalences cannot be achieved without changing other parameters: in the Two Normals model in particular, one cannot obtain a prevalence above 62% unless either the SD of the wider normal or the minimum effect of interest is changed.

peaks are similar to those in the dichotomous model (Fig 2B), as most effects are still close to 0 or 1, with negligible chances of crossing the minimum effect of interest threshold. We then proceed to an Overlapping Peaks model (Fig 1C), which uses a larger SD of 0.3 for Gaussians centered at 0 and 1 and leads to a larger overlap between distributions. This allows a greater overlap between normal centered at zero and one, leading to an effect size distribution that is more continuous, thus requiring arbitrary thresholds to differentiate between "true" and "false" effects. In this case, PPVs are slightly lower than in the two peaks model, particularly when power is high, as small effects under the minimum effect size of interest are eventually detected as positive. Nevertheless, the overall results are still quite similar to those of the previous models (Fig 2C).

Even though it is a more plausible scenario than the Dichotomous model, modeling effect sizes with peaks still seems unrealistic, as both the Two Peaks and Overlapping Peaks models are arbitrarily enriched in large effects around 1. A more plausible effect size distribution is a normal centered at zero, forming a continuous distribution where most effects sizes are small, and larger effects are found with a prevalence that varies with the standard deviation. Simulations in this case are initially performed with two normals centered at zero, with SDs of 0.1 and 1 (Two Normals model, Fig 1D). In this case, the prevalence of true effects is still adjusted by changing the relative weight of both normals while maintaining the SDs constant. This allowed for adjusting effect sizes and the prevalence of true effects independently; thus, large effects can still be observed even when most of the distribution is close to zero. We then move on to simulations using a Single Normal model (Fig 1E), in which the prevalence of true effects is controlled by varying the standard deviation alone. PPVs are smaller in these cases, as centering distributions around zero causes effects slightly under the minimum effect size of interest to be more frequent than those above it (Fig 2D and 2E). This in turn leads false-positive effects to be more common than when two peaks are used. Regardless of the effect size distribution, increasing power leads to a greater PPV, while higher bias decreases it.

In all of these scenarios, the decision to publish does not take into account the minimum effect size of interest–i.e. a significant result with a magnitude lower than 0.5 is still published. We do believe this is a realistic assumption in the biomedical sciences, in which sample size calculations or discussions about effect sizes tend to be scarce (e.g. [16]). That said, as shown in S3 Fig, including effect sizes in publishing decisions changes the results only when power is high, as estimates below the minimum effect of interest will not be significant if the power to detect this effect (which is smaller than those used for sample size calculations) is under 50%.

PPVs for the different models using the same parameter scenarios used in Ioannidis [1] are shown in Table 1. Results for the dichotomous model are very close to Ioannidis analytical predictions, confirming the validity of the simulations (for the actual prevalences obtained in each simulation, see S1 Table). For the Single Normal model, the PPV is lower than in the Dichotomous or Two Peaks distributions, as a normal centered at 0 has a higher mass of effects just below the minimum of interest. These effects will often be estimated above the critical value for significance and become false positives, especially when power is high, which makes them more likely to be detected. Interestingly, PPVs were inversely associated with the median p-value of published results (S4 Fig); thus, as proposed by Simonsohn and colleagues [17], analyzing p value distributions for published results in different scenarios provides a relatively straightforward way to estimate the prevalence of true effects in the literature (S5 Fig).

Importantly, the PPV for most of these scenarios is below 50% and falls markedly with decreasing prevalence of true effects and increasing bias. Even without bias, at a very low prevalence a high PPV can only be obtained with a combination of high power (low type II error rate) and a very stringent alpha (low type I error rate). The Ioannidis [1] model can thus be seen as a special case of our simulation framework (as approximated by the dichotomous

**Table 1. Positive predictive value (PPV) for simulated scenarios varying in power, bias and prevalence of true effects (i.e. above the minimum of interest).** Scenario descriptions, parameter combinations and the Analytical column are adapted directly from Ioannidis [1]. The remaining columns contain the PPVs for simulations of 5,000 published findings using each of the different effect size distributions in our model. Parameters for the distributions are set so that the prevalence of effects above the minimum of interest approximates the specified odds (see S1 Table for model parameters and achieved prevalences for each scenario). Missing values occur in some scenarios when the desired prevalence cannot be achieved without changing the minimum effect of interest, thus complicating direct comparisons with other scenarios. * For Two Normals, one cannot obtain a prevalence of 66.7% without changing the minimum effect of interest or the SD of the wider normal, as a normal N(0, 1) has only about 62% of its mass beyond the minimum of interest on both tails. ** For Two Peaks, one cannot obtain a prevalence of 0.1% without changing the minimum effect of interest, as a normal distribution N(0, 0.2) has over 1% of its mass beyond the minimum of interest on both tails.

| Scenario description | Power | Odds for True Effects | Prevalence of True Effects | Bias | Analytical | Dichotomous | Two Peaks | Overlapping Peaks | Two Normals | Single Normal |
|---|---|---|---|---|---|---|---|---|---|---|
| Adequately powered RCT with little bias and 1:1 pre-study odds | 80% | 1:1 | 50% | 10% | 85% | 85,0% | 84,4% | 79,7% | 79,3% | 76,0% |
| Confirmatory meta-analysis of good quality RCTs | 95% | 2:1 | 66,7% | 30% | 85% | 85,4% | 85,4% | 81,1% | - * | 80,5% |
| Meta-analysis of small inconclusive studies | 80% | 1:3 | 25% | 40% | 41% | 39,9% | 40,4% | 37,7% | 38,2% | 36,4% |
| Underpowered, but well-performed phase I/II RCT | 20% | 1:5 | 16,7% | 20% | 23% | 24,0% | 24,8% | 23,9% | 25,4% | 22,3% |
| Underpowered, poorly performed phase I/II RCT | 20% | 1:5 | 16,7% | 80% | 17% | 17,3% | 17,2% | 17,4% | 17,9% | 16,1% |
| Adequately powered exploratory epidemiological study | 80% | 1:10 | 9,1% | 30% | 20% | 19,9% | 19,6% | 17,3% | 19,0% | 16,8% |
| Underpowered exploratory epidemiological study | 20% | 1:10 | 9,1% | 30% | 12% | 12,5% | 11,7% | 12,0% | 13,6% | 12,1% |
| Discovery-oriented exploratory research with massive testing | 20% | 1:1000 | 0,1% | 80% | 0,10% | 0,1% | 0,1% | - ** | 0,3% | 0,1% |
| Discovery-oriented exploratory research with massive testing, but with more limited bias (more standardized) | 20% | 1:1000 | 0,1% | 20% | 0,15% | 0,2% | 0,2% | - ** | 0,2% | 0,1% |

model). Irrespective of how effect sizes are modeled, it is difficult to obtain a PPV above 50% when Ioannidis' assumptions hold true. In fields where publication bias and use of a fixed significance threshold of $\alpha = 0.05$ are prevalent, especially where effect sizes are small, most published research findings seem thus likely to be false in continuous effect size distributions as well.

There are caveats, of course, both in the assumptions and in the dichotomization between true and false hypotheses, which is necessary to calculate positive predictive values in a diagnostic screening framework. In our case, this was achieved by the use of an arbitrary minimum effect of interest threshold to define true studies. In reality, however, accurate estimation of effect sizes can be seen as more important than classifying effects as true or false on the basis of statistical significance [13, 18]. We thus also study how our modeled scenarios perform in terms of accuracy in estimating the magnitude of effect sizes.

Moving away from the idea that effects either exist or are practically zero, we analyze two other types of error in our simulations: sign (type S) errors and magnitude (type M) errors [14]. Sign errors occur when the estimated effect size has the opposite sign of the real effect size. Magnitude errors are measured by an exaggeration factor defined as the ratio between the estimated effect size and the true underlying effect, as overestimation is expected when publication decisions rely on statistical significance [19]. These errors are only measured in the simulations for published results–i.e. when a significant result is obtained. Of note, Type S and Type M errors cannot be defined if the real effect is exactly zero, and therefore are not evaluated in the Dichotomous model.

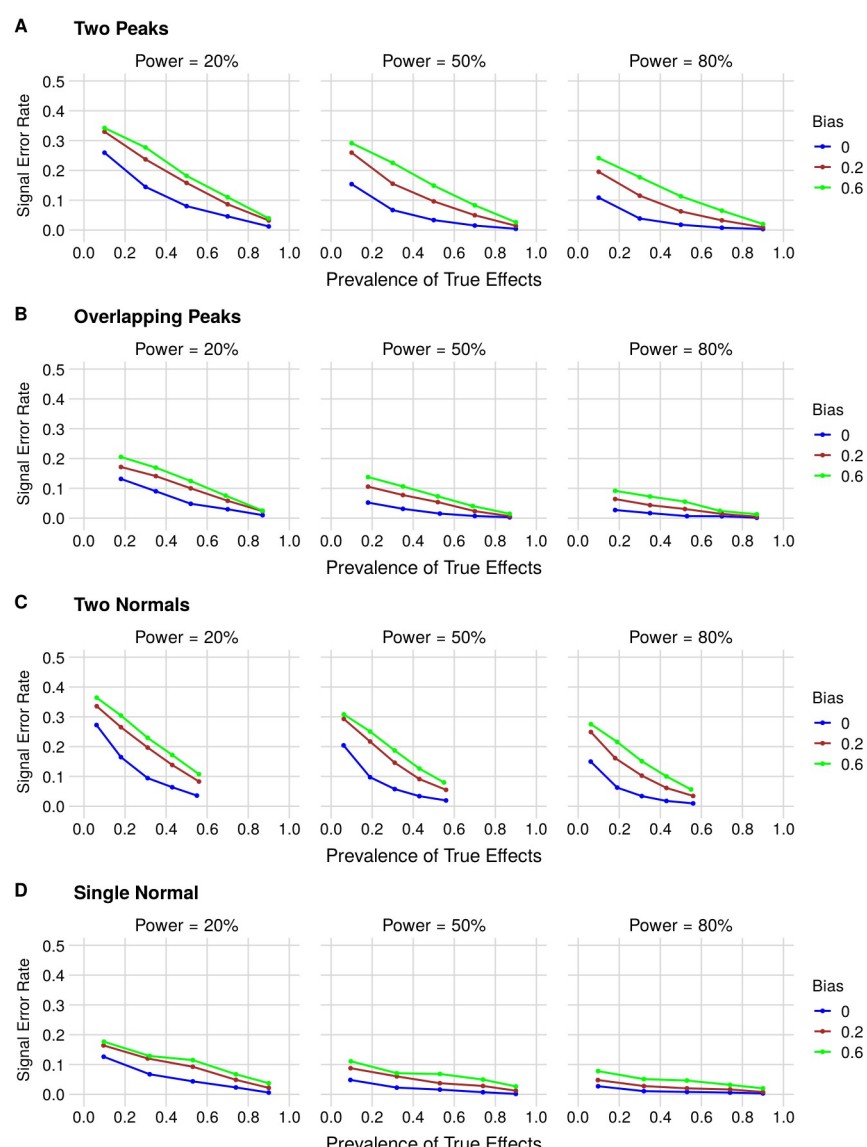

**Fig 3. Type S error rate (i.e. published estimates with the wrong sign) as a function of the prevalence of true effects, statistical power and bias for (A) Two Peaks, (B) Overlapping Peaks, (C) Two Normals and (D) Single Normal effect size distributions.** The rate corresponds to the percent of published (i.e. statistically significant) estimates that are in the opposite direction of the true effect. Each point corresponds to the sign error rate from a simulation with 5,000 published findings. In each panel, graphs correspond to 20%, 50% and 80% power from left to right. Blue, red and green curves correspond to bias of 0, 0.2 and 0.6, respectively. Prevalence ranges vary among scenarios, as for some distributions some prevalences cannot be achieved without changing other parameters.

The type S error rate for the other distributions is presented in Fig 3. Sign errors occur more frequently at lower power, as smaller sample sizes make it more likely that estimates will incorrectly yield significant results in the opposite direction. Bias also increases the sign error rate, as small effects, which are more likely to be estimated with the wrong sign, also become more likely to be published. Increasing the prevalence of true effects decreases the sign error rate, as this leads to a greater representation of large effects whose estimates are unlikely to cross zero. Thus, the error rate approaches 0 as the prevalence approaches 1, even in the presence of bias.

Interestingly, the frequency of type S errors is higher for distributions that are enriched in effects close to 0, as in the case of the Two Peaks and Two Normals model. For the Overlapping Peaks and Single Normal distributions, a higher fraction of false effects lies close to the minimum of interest–therefore, even though these might be incorrectly detected as positive results by effect size inflation, they are unlikely to be estimated with the wrong sign.

Similarly, the exaggeration of effect size estimates (i.e. type M error) is lower with increasing prevalence, increasing power and decreasing bias in all distributions. Increasing power produces better estimates, as larger sample sizes lead to lower variability between samples and smaller type M errors. On the contrary, when power is low, even true effects sometimes require exaggeration caused by sampling error to yield significant results, systematically inflating effect sizes in the literature [19]. Type M error also varies greatly depending on the prevalence of true hypotheses. Notably, most of the inflation comes from small effects that are inflated just beyond the significance threshold: as sampling error is fixed at 1, large errors are unlikely for large effects, while small effects near zero can result in very large exaggeration ratios. These findings mirror analytical results showing that low signal-to-noise ratios lead to high exaggeration [20].

Similarly to what is observed for type S errors, as the prevalence of large effects decreases, the higher proportion of small effects drives type M errors up: for the worst-case scenarios in which the false-positive rate is high and distributions are enriched in effects close to 0 (as in the Two Peaks and Two Normals models), the median exaggeration rate can reach values of up to 15. With high prevalence, on the other hand, this value is close to 1 for all models. These results illustrate that studying small effects with low power while publishing only significant findings can lead to a dramatic distortion of effect size estimations, as discussed by other authors [14, 17]. For type S error rate and median exaggeration factors for the specific scenarios proposed by Ioannidis [1], see S2 and S3 Tables.

Importantly, all results are dependent on the effect sizes used for power calculations. Our approach to base our calculations on a typical effect size (i.e. the mean of true effect sizes in a given distribution) means that, in the case of the Single Normal model, in which different prevalences are achieved by varying the SD, the effect size used for the calculations varies with prevalence, as larger prevalences imply larger mean effect sizes. To correct for this, we performed additional simulations using fixed values (either the minimum effect of interest or twice this value) for power calculations in this distribution. The results, shown on S6–S8 Figs, show that our results generally hold in this situation as well, and that using the higher value as a reference leads to even lower PPVs and greater type S and M error rates when prevalence is low.

## Discussion

Our results show that the overall conclusions of the dichotomized model presented by Ioannidis [1] generally hold for different distributions of continuous effect sizes as well. Depending on features of the literature, most published effect sizes might indeed be false positives—i.e. smaller than scientists would judge to be relevant for the field–when low statistical power, high bias and a low prevalence of large effects are combined. That said, using continuous effect sizes does result in a few differences from the original model. Our results for the specific scenarios in Ioannidis [1] show that, when power is high, the PPV is lower for continuous distributions than it is in the dichotomous model. This is due to the presence of small effects below the minimum of interest, that can nevertheless result in significant differences when power is high and count as false-positives. This exemplifies a trade-off between sensibility and specificity and shows that increasing power can have detrimental effects if statistical significance alone is used to judge results [13].

Interestingly, when the focus is switched from classifying effects as true or false to accurately estimating the direction and magnitude of effects, these conclusions are reversed: sign error rates and effect size inflation (i.e. type S and M errors) are less problematic in continuous distributions than in dichotomous ones in which a larger fraction of effects are close to 0. This illustrates a limitation of the diagnostic screening framework, as there is information in an experiment that goes beyond classifying a hypothesis as true or false and will not be captured by the PPV. That said, the combination of low prevalence of true hypotheses, low statistical power and high bias has a detrimental effect both on the PPV and on type S and M errors, once again suggesting that the main conclusions drawn by Ioannidis [1] hold true in an estimation framework as well.

One way of seeing the results of such models is to consider that science has two steps: one of signal detection, where using significance testing as modeled here is more appropriate, and a second phase of estimation and understanding, when statistical significance alone cannot provide accurate guidance [13, 21]. Within this logic, the problem is when both approaches are merged into a single step and the results of "screening science" are taken as more than provisional. Our model shows that this practice not only leads to false-positive results, but also to serious errors in estimation, as the rates of sign and magnitude errors are also affected by statistical power and bias. Even in the absence of bias, median effect sizes can be markedly exaggerated if studies are underpowered, reinforcing commonly voiced concerns about sample size calculations using estimates from the literature [13, 14]. It also highlights the dangers of the uncritical use of NHSTs, which have been repeatedly pointed out by various authors [4, 13, 22].

Although our simulations address what is perhaps the main methodological shortcoming of the diagnostic screening framework used by Ioannidis [1], other limitations of the original model still apply to our work. Particularly, its assumptions of absolute publication bias, lack of replication and publication of significant findings irrespective of context are quite pessimistic: the model assumes that scientists will follow the rules of NHSTs blindly, exclusively and invariably publishing significant results with no consideration of effect sizes, prior probabilities or other findings. Although this is certainly an exaggeration, we do believe that excessive reliance on significance tests happens frequently enough for this approximation to be reasonable [23].

Nevertheless, other results have shown that publication of negative results and replication of published findings can do a lot to mitigate the effect of bias and low power [8, 9, 24]. It is also worth pointing out that, as statistical power and risk of bias can be assessed–even if imperfectly–from the literature, an assessment of these features in the published literature can help select for studies with higher positive predictive values. In this sense, meta-analyses and other forms of data synthesis which can take account both the weight and risk of bias of included studies can plausibly generate more reliable results than individual studies.

In our model, effect sizes were determined by mean differences between groups in two sample t-tests; nevertheless, similar results would probably be obtained for experiments with correlations or other effect size measures, as standardized effect sizes are generally interconvertible. Another limitation in our framework is the fact that it requires one to set arbitrary minimum effect sizes of interest to correspond to the notion of true and false hypotheses in the diagnostic framework terminology. Although we concede that such dichotomies are artificial, they are needed in order to maintain correspondence with previous models–and with dichotomous views that are still widespread in many areas of science [25]. Nevertheless, the dissociation between type S and type M error rates and false-positive rates for different distributions illustrates the limitations of using any of these approaches in isolation. An additional limitation is that, for continuous models, varying prevalence inevitably requires adjustments in other parameters in the effect size distributions. Thus, prevalence cannot be adjusted independently, and results can be influenced by changes in other parameters.

Concerning more philosophical issues, the difficulties of interpreting the *a priori* prevalence of true hypotheses in the Ioannidis model [6, 4, 13] are replaced by concerns about the shape of the underlying distribution of effect sizes in our model. We are partial to arguments made for distributions of the exponential family for effect sizes, centered near zero, with smaller effects being more common than large ones (e.g. [13]). Nevertheless, we chose to experiment with different distributions to test the sensitivity of the model–from an unrealistically dichotomous one to a continuous distribution with most effects around 0. In spite of the chosen distribution, the general conclusions concerning false-positive rates in different scenarios and their sensitivity to power, bias and the prevalence of true effects seem to hold, albeit with small variations. Nevertheless, even among continuous distributions with exponential tails (e.g. Two Normals and Single Normal), some parameters such as the median effect size inflation in the literature were quite sensitive to the exact shape of the curve, suggesting that estimates from simulations should be taken with caution in the absence of empirical evidence on effect size distributions. Such evidence can eventually be obtained from large meta-analyses or replication initiatives, but will typically be limited by publication bias and thus inevitably consist in a rough approximation.

As future steps, we plan to extend the model to have negative results being published at different rates, thus relaxing the full publication bias assumption. It will also be interesting to include replications and interlaboratory variation, in order to explore how different criteria for replication track the features of the actual effects, whether meta-analyses to aggregate results have higher reliability than individual studies and whether the results of systematic replications can be used to provide empirical estimates of statistical power, bias and prevalence of true hypotheses within a scientific field. This will serve the purpose of informing the development of the analysis plan for the Brazilian Reproducibility Initiative, a multicenter replication of Brazilian biomedical science which we currently coordinate [26], and hopefully provide a reference for interpreting results of large-scale replications in general [27].

## Methods

### Model overview

In our model, scientists pick effects to be tested at random from an underlying distribution of true effect sizes, each of which represents the difference between the population means of a quantitative variable in two groups. Although expressing effect sizes as differences is a common situation in basic biomedical research, they could also be measured in other forms (e.g. correlation coefficients) for similar purposes. Scientists then obtain a sample from each experimental group and compare their means. If the difference is statistically significant, the results are published. If they are not, the result will not be published unless bias is present. The bias parameter dictates the proportion of negative results that artificially reach significance through biased analysis or other means, as in Ioannidis [1]. After a predefined number of findings are published, we can evaluate the accuracy of the literature produced under a given condition by different metrics. We describe the steps and parameters in more detail below.

### Effects

For each iteration of the simulation, an effect is chosen at random from an underlying distribution that represents the effect sizes investigated by scientists within a research field. The effect size distribution in a given field cannot be known in advance, and unbiased data are scarce to provide good estimates. Literature overviews (e.g. [16, 28]) and systematic replication initiatives (e.g. [29]) can provide some insights on the subject, but are limited by publication bias in the former case and by small numbers of studies in the latter. Nevertheless, arguments

**Table 2. Model parameters for different effect size distributions.** Columns show parameters used for each version of the model. The weight of the second normal $W_B$ is varied to determine prevalence for the different simulations. When a single normal is used, this is accomplished by varying $SD_A$. For Figs 2–4, $W_B$ was set to 0.1, 0.3, 0.5, 0.7 and 0.9 to achieve prevalences between 0.1 and 0.9 in the first 4 models. In the Single Normal model, $SD_A$ was set to 0.3, 0.5, 0.8, 1.5 and 4 to approximate the same prevalences.

| Scenario | $SD_A$ | $\mu_B$ | $SD_B$ | $W_B$ |
|---|---|---|---|---|
| **Dichotomous** | 0 | 1 | 0 | varies |
| **Two Peaks** | 0.1 | 1 | 0.1 | varies |
| **Overlapping Peaks** | 0.3 | 1 | 0.3 | varies |
| **Two Normals** | 0.1 | 0 | 1 | varies |
| **Single Normal** | varies | 0 | 0 | 0 |

have been made that exponential curves or other distributions with most of their density around 0 might be a good approximation, as most effects are likely to be small, with larger effect sizes becoming progressively rarer [13].

We parametrize the effect size distribution as a mixture of two normal distributions. The first one, distribution A, is centered in 0, with standard deviation $SD_A$. The second one, distribution B, is centered at $\mu_B$, with standard deviation $SD_B$. The relative weight of distribution B in the mixture is given by $W_B$, between 0 and 1. From this parametrization, we define 5 types of distributions to model possible scenarios in different scientific fields (see Fig 1 for illustration and Table 2 for parameters).

The *Dichotomous* model is equivalent to that in Ioannidis [1], where effects are dichotomized as true/false. Here, effects are either large or null–exactly 1 or exactly 0 –, approximating the Ioannidis [1] model, in which the fact that power is fixed implicitly assumes that true effect sizes have similar magnitudes. The *Two Peaks* model is a more realistic dichotomous model, with Gaussian variation around means of 1 and 0 for true and null effects, respectively–nevertheless, it still yields a clear distinction between both categories of effects. The *Overlapping Peaks* has peaks with a broader spread, with the two normal distributions showing a much greater degree of overlap–which effectively precludes defining them as containing "null" and "true" effects. In the *Two Normals* model, both normals are centered at 0 and differ only in their standard deviation. Our last scenario uses a *Single Normal* as the underlying distribution of effect sizes for simplicity and plausibility.

As effect sizes are modeled as continuous, calculating measures such as the false positive rate or the positive predictive value requires defining a minimum effect of interest to determine what constitutes a true effect–i.e. one whose absolute size is larger than the minimum. This parameter is also used for statistical power/sample size calculations (see below) and is set to 0.5 for all simulations. As the standard deviation for the samples is fixed at 1 (see below), effect sizes can be interpreted as Cohen's *d*, with our minimum corresponding to a medium effect size in Cohen's original classification [30].

To allow the prevalence of true hypotheses—a key parameter in the Ioannidis [1] model —to vary in models with two distributions, we change the relative weight of both distributions ($W_B$) while maintaining the other parameters fixed. For the Single Normal model, in which $W_B$ is set to 0, variation in the prevalence of true effects is achieved by changing the standard deviation, $SD_A$ (see Table 2). Importantly, this causes the range of achievable prevalences to vary from model to model, precluding simulation of some of the scenarios in Table 1 and S1–S3 Tables. The absolute values of the minimum of interest and the mean of normal distributions are arbitrary and should not influence results by themselves, as long as statistical power and the prevalence of "true" effects are calculated relative to the minimum.

## Sampling

Once an effect size **ES** is investigated by a scientist, experimental samples are generated. For the control group, a sample of a fixed size is drawn from a normal distribution N(0,1). For the treatment group, a sample of the same size is drawn from a normal N(**ES**,1). Sample size is determined by setting statistical power to the desired value. Power is specified to detect the average of the effect sizes above the minimum of interest within the model's distribution of effects (by sampling 100,000 effects from the distribution under study). This can be thought as a "typical" true effect size in a given research field, which could plausibly be used as a reference by researchers on the basis of the published literature. This means that in the Dichotomous and Two Peaks scenarios, statistical power refers to a difference very close to 1. In the Single Normal model, as prevalence increases, the larger SD (up to 4) means that typical effect sizes become larger; therefore, power for a given effect size decreases as prevalence increases. To control for this fact, we also run simulations changing the minimum of interest for power calculations for this particular model in S6 and S7 Figs.

Effects above 0.5 (i.e. those we call true effects) lead to true-positive results when the observed effect is both significant and in the same direction as the real one, or to false negatives when a non-significant result is obtained. Effects below 0.5 (i.e. null effects) lead to true-negative results when the observed effect is non-significant or to false positives when a significant result is obtained in either direction, either through chance or bias. In cases where a significant estimate with the opposite sign of the actual effect is obtained, it is counted as a false positive.

## Analysis and publication of results

After sampling, a two-tailed Student's t test is performed to compare the two samples, assuming equal variances. If the result indicates a statistically significant difference at $\alpha = 0.05$, the finding is published as the estimated difference between the means of the two groups. If the test does not return a significant result, one of two outcomes happens depending on bias probability **b**: either the result is not published with probability 1–**b**, or the experiment is repeated until the critical value for significance is reached in either direction and then published with probability **b**. When the bias term leads to repetition of experiments, only the final significant result is considered and published. Both the assumption of publication bias and the implementation of bias represent simplified and pessimistic versions of the actual scientific process. Nevertheless, the bias term provides a simple way to model the outcome of both misconduct and questionable research practices, such as HARKing and flexibility of analysis, under the assumption that, with enough degrees of freedom, a significant p-value should be easy to find [15]. Thus, although the number of repetitions would likely be limited patience and resources in real life, high bias could still be obtained by these other means, as well as by increasing the propensity to repeat experiments.

Notice that the decision to publish in the main simulations does not take into account the minimum effect of interest–i.e. a significant result with a magnitude lower than 0.5 is still published. We believe this is a realistic assumption in basic biomedical science: as most articles do not present sample size calculations or discussions about effect sizes [31], it seems reasonable to assume that they do not influence the decision to publish. Nevertheless, we include simulations where results are only published if they are significant and larger than the minimum of interest, which are presented in S3 Fig.

## Evaluation of the generated literature

Reported simulations consist of 5,000 published findings for each combination of parameters. For these findings, we compute different measures of accuracy: the positive predictive value

(PPV) the type S error rate and the median type M error magnitude [14]. We also analyze the distribution of p-values found among published results for each scenario (S4 and S5 Figs).

The PPV (Fig 2 and Table 1) is calculated as the percentage of positive findings in the literature that represent true positives, as defined by the minimum effect of interest. Since we assume complete publication bias, every published result is positive. Thus, true effects are defined as those whose true absolute value is larger than the minimum effect size of interest. Once again, we note that true positives are only counted as such if they have been estimated to have the same sign as the real effect.

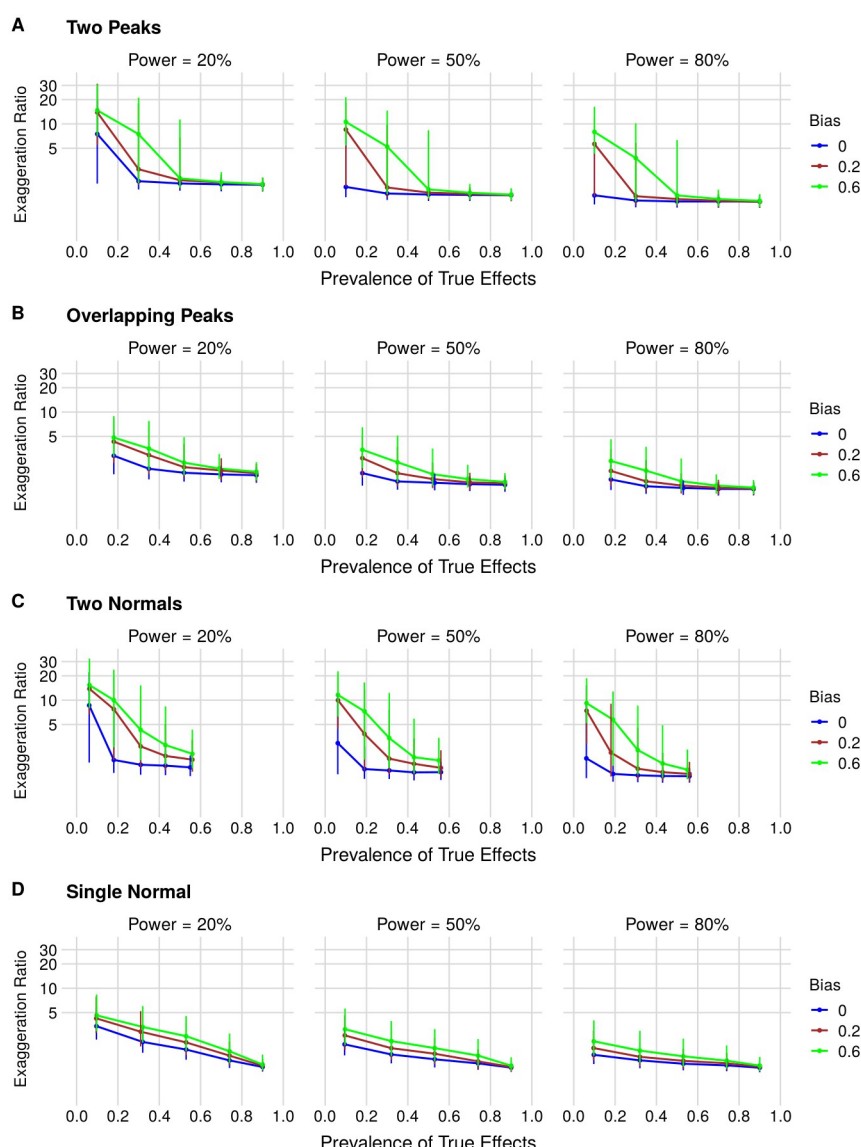

**Fig 4. Type M error (i.e. effect size exaggeration) as a function of the prevalence of true effects, statistical power and bias for (A) Two Peaks, (B) Overlapping Peaks, (C) Two Normals and (D) Single Normal effect size distributions.** Each point corresponds to the median exaggeration ratio for 5,000 published effects in the simulated literature, with error bars indicating the interquartile range. The Y axis represents the exaggeration ratio in a log scale to allow better visualization. In each panel, graphs correspond to 20%, 50% and 80% power from left to right. Blue, red and green curves correspond to bias of 0, 0.2 and 0.6, respectively. Prevalence ranges vary among scenarios, as for some distributions some prevalences cannot be achieved without changing other parameters.

The type S error rate (Fig 3) is the proportion of significant estimates with the opposite sign of the real effect in the published literature, irrespective of them being above or below the minimum effect size of interest. Note that, as full publication bias is assumed, this will only happen when a significant estimate with the opposite sign is obtained. For published estimates that have the correct sign, we also compute the type M error (Fig 4), or exaggeration factor, by dividing the obtained estimate by the true effect size for each simulation. For each parameter combination, we report the median and interquartile range of the exaggeration factor for the simulated published literature.

## Source code and data

The model was developed in R 3.6.3 [32]. Code for the model, along with auxiliary scripts used to make graphs and analyses, are available on GitHub (https://github.com/KleberNeves/reproducibility-model). The generated data for the simulations shown are available at https://osf.io/z2hn9/files/. The model is available as a Shiny App at https://kneves.shinyapps.io/repro-model/, where one can change the parameters described above (as well as others), run simulations and explore the results.

## Supporting information

**S1 Fig. A diagram summarizing each iteration of the simulations (described in detail in the Methods section).** Given an effect size drawn from the distributions described in Fig 1, samples are generated for two groups, from normal probability distributions whose means differ by the effect size. A t-test is performed comparing these two samples and a p-value is obtained. If that p-value is significant at the given alpha level, the result is published. If it is not, there is a chance that the result will be biased, with a probability given by the bias parameter. If it is, new samples will be generated until significance is reached and the result is published. Otherwise, the result is not published.
(TIF)

**S2 Fig. PPV as a function of the odds of true effects, statistical power and bias for the Dichotomous effect size distribution.** These are the same results shown in Fig 2A(capped at an odds ratio of 1), and are meant to be more directly comparable to the figures in Ioannidis [1], which specify prevalence using odds instead of percentages. Each point corresponds to a simulation of 5,000 published findings. True effects are defined as those above the minimum of interest (i.e. Cohen's d> 0.5 in all simulations). In each panel, graphs correspond to 20%, 50% and 80% power from left to right. Blue, red and green curves correspond to bias of 0, 0.2 and 0.6, respectively.
(TIF)

**S3 Fig. Relationship between positive predictive value and prevalence of true effects for (A) Dichotomous, (B)Two Peaks, (C)Overlapping Peaks, (D)Two Normals and (E) Single Normal effect size distributions.** These results are similar to those presented in Fig 2, except that the decision to publish takes the estimated effect size into account–i.e. results are only published if the estimate was larger than the minimum of interest. Each point corresponds to a simulation with 1,000 published findings. True effects are defined as those above the minimum of interest (i.e. Cohen's d> 0.5 for these simulations). In each panel, graphs correspond to 20%, 50% and 80% power from left to right. Blue, red and green curves correspond to bias of 0, 0.2 and 0.6, respectively. Prevalence ranges vary among scenarios, as for some distributions some prevalences cannot be achieved without changing other parameters.
(TIF)

**S4 Fig. Relationship between the median p-value of published results and prevalence of true effects for (A) Dichotomous, (B)Two Peaks, (C)Overlapping Peaks, (D)Two Normals and (E) Single Normal effect size distributions.** Each point corresponds to a simulation with 1,000 published findings. In each panel, graphs correspond to 20%, 50% and 80% power from left to right. Blue, red and green curves correspond to bias of 0, 0.2 and 0.6, respectively. Prevalence ranges vary among scenarios, as for some distributions some prevalences cannot be achieved without changing other parameters.
(TIF)

**S5 Fig. Examples of p-curves obtained for different scenarios.** Plots show histograms of the distribution of p-values for published (i.e. significant) results for 1,000 findings, in different scenarios. Top panels show results for the Two Peaks model, and bottom ones for the Single Normal model. Left panels show results for a high-bias, low-power, low-prevalence scenario, while right ones have no bias, high power and high prevalence. Specific parameters are (A) Two Peaks, bias = 0.6, power = 20%, prevalence ~ 0.1,PPV ~ 11%(B)Two Peaks, bias = 0, power = 80%, prevalence ~ 0.9, PPV ~ 12%, (C)Single Normal, bias = 0.6, power = 0.2, prevalence ~ 0.1, PPV ~ 99% and (D)Single Normal, bias = 0, power = 0.8, prevalence ~ 0.9, PPV ~ 99%.
(TIF)

**S6 Fig. PPV as a function of the prevalence of true effects, statistical power and bias for the single normal model when different effect sizes are used for power calculations.** In (A), sample size is calculated to detect the mean of the true effect sizes above the minimum of interest (i.e. Cohen's d > 0.5). This is the method used in the main simulations. As prevalence in the Single Normal model is increased by increasing the SD, the mean effect size above the minimum is larger as prevalence increases, leading simulations to be underpowered compared to other distributions at high prevalences. In (B), sample size is calculated to detect an effect equal to twice the minimum of interest (i.e. Cohen's d = 1). In (C), it is calculated to detect an effect equal to the minimum of interest (i.e. Cohen's d = 0.5). Each point corresponds to a simulation with 1,000 published findings. True effects are defined as those above the minimum of interest. In each panel, graphs correspond to 20%, 50% and 80% power from left to right. Blue, red and green curves correspond to bias of 0, 0.2 and 0.6, respectively.
(TIF)

**S7 Fig. Type S error rate (i.e. published estimates with the wrong sign) as a function of the prevalence of true effects, statistical power and bias when different effect sizes are used for power calculations.** In (A), sample size is calculated to detect the mean of the true effect sizes above the minimum of interest (i.e. Cohen's d > 0.5). This is the method used in the main simulations. As prevalence in the Single Normal model is increased by increasing the SD, the mean effect size above the minimum is larger as prevalence increases, leading simulations to be underpowered compared to other distributions at high prevalences. In (B), sample size is calculated to detect an effect equal to twice the minimum of interest (i.e. Cohen's d = 1). In (C), it is calculated to detect an effect equal to the minimum of interest (i.e. Cohen's d = 0.5). Each point corresponds to a simulation with 1,000 published findings. True effects are defined as those above the minimum of interest. In each panel, graphs correspond to 20%, 50% and 80% power from left to right. Blue, red and green curves correspond to bias of 0, 0.2 and 0.6, respectively.
(TIF)

**S8 Fig. Type M error (i.e. median effect size exaggeration) as a function of the prevalence of true effects, statistical power and bias.** In (A), sample size is calculated to detect the mean of the true effect sizes above the minimum of interest (i.e. Cohen's d > 0.5). This is the method

used in the main simulations. As prevalence in the Single Normal model is increased by increasing the SD, the mean effect size above the minimum is larger as prevalence increases, leading simulations to be underpowered compared to other distributions at high prevalences. In (B), sample size is calculated to detect an effect equal to twice the minimum of interest (i.e. Cohen's d = 1). In (C), it is calculated to detect an effect equal to the minimum of interest (i.e. Cohen's d = 0.5). Each point corresponds to a simulation with 1,000 published findings. True effects are defined as those above the minimum of interest. In each panel, graphs correspond to 20%, 50% and 80% power from left to right. Blue, red and green curves correspond to bias of 0, 0.2 and 0.6, respectively.
(TIF)

**S1 Table. Actual prevalences of true effects obtained in the simulations for the simulated scenarios in Table 1.** Scenario descriptions and parameter combinations are adapted directly from Ioannidis [1]. The remaining columns contain the results from simulations of 5,000 published findings using different effect size distributions in our model. The parameters for the distributions are set so that the prevalence of effects above the minimum effect of interest approximate the odds specified by the scenario. Missing values in some scenarios occur when the desired prevalence cannot be achieved without changing the minimum effect of interest, thus complicating direct comparisons with other scenarios. * For Two Normals, one cannot obtain the desired prevalence (66,7%) without changing the minimum of interest threshold or the SD of the wider normal. A normal N(0, 1) has only about 62% of its mass beyond the minimum of interest. ** For Two Peaks, one cannot obtain the desired prevalence (0,1%) without changing the minimum of interest threshold. A normal distribution N(0, 0.2) -has over 1% of its mass beyond the minimum of interest on both sides.
(XLSX)

**S2 Table. Type S error rates for the simulated scenarios in Table 1.** Scenario descriptions and parameter combinations are adapted directly from Ioannidis [1]. The remaining columns contain the prevalence of sign errors in simulations of 5,000 published findings using different effect size distributions in our model. The parameters for the distributions are set so that the prevalence of effects above the minimum effect of interest approximate the specified odds (see S1 Table for model parameters and achieved prevalences for each scenario). Missing values in some scenarios occur when the desired prevalence cannot be achieved without changing the minimum effect of interest, thus complicating direct comparisons with other scenarios.* For Two Normals, one cannot obtain the desired prevalence (66,7%) without changing the minimum of interest threshold or the SD of the wider normal. A normal N(0, 1) has only about 62% of its mass beyond the minimum of interest. ** For Two Peaks, one cannot obtain the desired prevalence (0,1%) without changing the minimum of interest threshold. A normal distribution N(0, 0.2) -has over 1% ofits mass beyond the minimum of interest on both sides.
(XLSX)

**S3 Table. Median exaggeration factor for the simulated scenarios in Table 1.** Scenario descriptions and parameter combinations are adapted directly from Ioannidis [1]. The remaining columns contain the median effect size exaggeration in simulations of 5,000 published findings using different effect size distributions in our model. The parameters for the distributions are set so that the prevalence of effects above the minimum effect of interest approximate the specified odds (see S1 Table for model parameters and achieved prevalences for each scenario). Missing values in some scenarios occur when the desired prevalence cannot be achieved without changing the minimum effect of interest, thus complicating direct comparisons with other scenarios.* For Two Normals, one cannot obtain the desired

prevalence (66,7%) without changing the minimum of interest threshold or the SD of the wider normal. A normal N(0, 1) has only around 62% of its mass beyond the minimum of interest. ** For Two Peaks, onecannot obtain the desired prevalence (0,1%) without changing the minimum of interest threshold. A normal distribution N(0, 0.2) -has over 1% of its mass beyond the minimum of interest on both sides.
(XLSX)

## Author Contributions

**Conceptualization:** Kleber Neves, Pedro B. Tan, Olavo B. Amaral.

**Formal analysis:** Kleber Neves, Pedro B. Tan, Olavo B. Amaral.

**Software:** Kleber Neves.

**Visualization:** Kleber Neves.

**Writing – original draft:** Kleber Neves, Pedro B. Tan, Olavo B. Amaral.

**Writing – review & editing:** Kleber Neves, Pedro B. Tan, Olavo B. Amaral.

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
