## [Decision Letter · Decision Letter 0]

29 Mar 2022

PONE-D-22-00486Are most published research findings false in a continuous universe?PLOS ONE

Dear Dr. Neves,

Thank you for submitting your manuscript to PLOS ONE. After careful consideration, we feel that it has merit but does not fully meet PLOS ONE’s publication criteria as it currently stands. Therefore, we invite you to submit a revised version of the manuscript that addresses the points raised during the review process. Please submit your revised manuscript by May 13 2022 11:59PM. If you will need more time than this to complete your revisions, please reply to this message or contact the journal office at plosone@plos.org. Please include the following items when submitting your revised manuscript:A rebuttal letter that responds to each point raised by the academic editor and reviewer(s). You should upload this letter as a separate file labeled 'Response to Reviewers'.A marked-up copy of your manuscript that highlights changes made to the original version. You should upload this as a separate file labeled 'Revised Manuscript with Track Changes'.An unmarked version of your revised paper without tracked changes. You should upload this as a separate file labeled 'Manuscript'.

We look forward to receiving your revised manuscript.

Kind regards,

Antonio Calcagnì, Ph.D.

Academic Editor

PLOS ONE

Journal Requirements:

“This work was supported through grants from FAPERJ (E-26/203.222/2017; OBA) and the Serrapilheira Institute (OBA; KN; PBT).”

“This work was supported through grants from FAPERJ (E-26/203.222/2017; OBA) and the Serrapilheira Institute (OBA). The funders had no role in study design, data collection and analysis, decision to publish, or preparation of the manuscript.”

Reviewer's Responses to Questions

**Comments to the Author**

1. Is the manuscript technically sound, and do the data support the conclusions?

Reviewer #1: Yes

Reviewer #2: Yes

2. Has the statistical analysis been performed appropriately and rigorously? 

Reviewer #1: Yes

Reviewer #2: Yes

3. Have the authors made all data underlying the findings in their manuscript fully available?

Reviewer #1: Yes

Reviewer #2: Yes

4. Is the manuscript presented in an intelligible fashion and written in standard English?

Reviewer #1: Yes

Reviewer #2: Yes

5. Review Comments to the Author

Reviewer #1: The manuscript presented a simulation framework as an improvement of Ioannidis’s (2005) work where hypothetical scientific literature is simulated from underlying continuous distributions of effect sizes, allowing the extraction of several parameters such as the positive predictive value and type-M/S errors. In addition, the authors provided a well-designed Shiny application to perform the simulation more intuitively. I found the simulation approach very interesting and flexible, especially for the possibility to include more parameters (such as the interlaboratory variation). Furthermore, implementing magnitude and direction errors (type-M/S) allows a clear and complete overview of the simulated literature.

As a practical note, in my comments, I am referring to the pdf page, given that the page number is missing.

Main points

1.Through the manuscript, it is not clear the empirical relevance of different hypothetical effect size distributions. Referring to Figure 1, the distributions B, C, D, and E are differentiated by Ioannidis’ s (2005) model, highlighting the implausibility of binary true/false effects. However, the empirical meaning of other distributions is less clear. I think that authors should expand more on the difference between different scenarios. For example, the “Two Peaks” model and the “Overlapping Peaks” model are essentially two instances of the same parametrization (lower or greater standard deviation) but here they are presented essentially as different models, without appropriate explanation. Similarly, while the “Two Normals’‘ model's parametrization is different from the “Single Normal” model, the empirical difference is not clear.

2. I think that the paragraph (p. 14) “Importantly, the PPV for most of these scenarios is below 50% and falls markedly with decreasing prevalence of true effects and increasing bias […]” is very important and should be expanded. The simulations support Ioannidis’s conclusions under specific conditions. In fact, Ioannidis’ s(2005) approach can be considered as a special case of the proposed simulation framework and this is a crucial point of the paper.

3. Given that estimation of the real effect size distribution is the main aim of the random-effects meta-analysis, I think that it is important to discuss the role of meta-analysis within this simulation framework. For example, can the simulation approach be improved by meta-analytic estimates of the average effect and standard deviation (especially for the “Single Normal” model)? Furthermore, can a meta-analysis benefit from this simulation framework?

4. In general, I found some explanations of simulation parameters not completely clear. In particular:

a. In the Method section (Model overview subsection) it is unclear how the bias parameter is formalized. Given that

it is a crucial element of the simulation approach, especially compared to Ioannidis’s (2005) work. When an

experiment is not significant, there is a probability of 1 - b of not publishing. Does this mean that when a non-

significant experiment is repeated with probability b, the significant result (after repetitions) is then considered?

The authors should expand the Bias parameter description and implementation within the simulation.

b. Why did the authors always fix to 1 the mean of the distributions (except for the “Two Normals” and “Single

Normal” models) and why is the minimum effect size of interest set to 0.5? For example, in the social sciences

(e.g., Social Psychology) the average effect size is estimated at around 0.3 and a 0.5 as a minimum effect size of

interest seems quite implausible.

5. Related to the previous point, I suggest the inclusion of a simulation flowchart diagram to highlight all relevant steps and parameters. This will improve the manuscript’s readability.

6. In the Sampling section (p.22) is not clear how the author set the sample size based on the desired power level (“Power is specified to detect a typical difference for the field, defined as the average of the effect sizes above the minimum of interest within the model’s distribution of effects (by sampling 100,000 effects from the distribution under study)”). Looking at the source code, the power is analytically calculated using the power.t.test() function but which true effect is used within the function? Could the authors provide a clearer explanation of this step?

7. In the Discussion (p.19), the authors point out that simulations’ results are strongly influenced by the assumed effect size distribution (“parameters such as the median effect size inflation in the literature were quite sensitive to the exact shape of the curve, suggesting that estimates from simulations should be taken with caution in the absence of empirical evidence on effect size distributions”). At the same time, it is difficult to estimate the true effect size distribution from published data, especially under some conditions. How do authors suggest dealing with this point? s it possible to quantify the impact of the distribution choice on simulations’ results?

Minor points

1. I think that Figure 1 could be improved by plotting also the area under the curve according to true/false positive/negative probabilities (for example using transparency for overlapping curves)

2. The author used the term “signal” in the context of the effect size direction (type S error). I think that it is more appropriate to use the term “sign” as in the mathematical sense.

3. The simulation’s assumption that results are published even if the effect size is lower than the minimum effect of interest is quite reasonable. I think that an interesting scenario could be to simulate a different publication criterion. Do the authors consider some scenarios where publishing is based on the minimum effect size of interest instead of p-values only?

4. The Github repository contains another simulation parameter called Negative Results Incentive defined as the probability to publish a non-significant result. I would suggest including this parameter in the paper (e.g., in the future directions section) given the current relevance of publishing null findings.

Further comments (not required for revision)

I appreciate the effort of organizing and sharing the source code for the Shiny app and the manuscript simulation reproducibility. However, I would suggest improving the repository organization and documentation because I have found it a little bit difficult to follow the simulation steps (especially what is not clear from the manuscript). While the Shiny app is well constructed, the source code is sometimes difficult to understand. Some suggestions:

- A unique documentation file/s where the simulation and folders organization is clearly explained

- More comments on created functions to understand parameters and computations.

Reviewer #2: I read and appreciated the article.

Although the results are quite predictable, I think it is important to highlight them.

In this kind of works, it is often necessary to hypersemplify the reality, but I think that the chosen scenarios are quite representative.

I also tried the shiny-app and found it very interesting.

Consequently I don't have particular comments.

Minor points:

- I would have appreciated page numbers in the draft

- page 21 "From this parametrization, we define 5 types of distributions to model

possible scenarios in different scientific fields (see Figure 2 for illustration and Table 2 for

parameters)." It Seems to me that this refers to Figure 1 instead.

6. PLOS authors have the option to publish the peer review history of their article (what does this mean?). If published, this will include your full peer review and any attached files.

Reviewer #1: No

Reviewer #2: No

---

## [Author Response · Author response to Decision Letter 0]

14 Sep 2022

Reviewer #1: The manuscript presented a simulation framework as an improvement of Ioannidis’s (2005) work where hypothetical scientific literature is simulated from underlying continuous distributions of effect sizes, allowing the extraction of several parameters such as the positive predictive value and type-M/S errors. In addition, the authors provided a well-designed Shiny application to perform the simulation more intuitively. I found the simulation approach very interesting and flexible, especially for the possibility to include more parameters (such as the interlaboratory variation). Furthermore, implementing magnitude and direction errors (type-M/S) allows a clear and complete overview of the simulated literature.

As a practical note, in my comments, I am referring to the pdf page, given that the page number is missing.

> We have added page numbers to the manuscript (and apologize for not having done this in the first version). We use this numbering to refer to specific passages in our responses.

Main points

1.Through the manuscript, it is not clear the empirical relevance of different hypothetical effect size distributions. Referring to Figure 1, the distributions B, C, D, and E are differentiated by Ioannidis’ s (2005) model, highlighting the implausibility of binary true/false effects. However, the empirical meaning of other distributions is less clear. I think that authors should expand more on the difference between different scenarios. For example, the “Two Peaks” model and the “Overlapping Peaks” model are essentially two instances of the same parametrization (lower or greater standard deviation) but here they are presented essentially as different models, without appropriate explanation. Similarly, while the “Two Normals’‘ model's parametrization is different from the “Single Normal” model, the empirical difference is not clear.

> A brief rationale for these modelling choices was introduced in the text (pages 4 and 5). That said, we do stress that we do not have good empirical data on effect size distributions, as even works that have attempted to document this in certain areas (e.g. Szucs and Ioannidis, 2017 (http://doi.org/10.1371/journal.pbio.2000797), Carneiro et al., 2018 https://doi.org/10.1371/journal.pone.0196258 are based on the published literature and have no way to account for publication bias (as added on page 14).

2. I think that the paragraph (p. 14) “Importantly, the PPV for most of these scenarios is below 50% and falls markedly with decreasing prevalence of true effects and increasing bias […]” is very important and should be expanded. The simulations support Ioannidis’s conclusions under specific conditions. In fact, Ioannidis’s (2005) approach can be considered as a special case of the proposed simulation framework and this is a crucial point of the paper.

> This has been made explicit in the text, when describing the model from Ioannidis and comparing results with continuous scenarios (page 8).

3. Given that estimation of the real effect size distribution is the main aim of the random-effects meta-analysis, I think that it is important to discuss the role of meta-analysis within this simulation framework. For example, can the simulation approach be improved by meta-analytic estimates of the average effect and standard deviation (especially for the “Single Normal” model)? Furthermore, can a meta-analysis benefit from this simulation framework?

> Implementing meta-analyses was beyond the scope of this project as, for this set of results in particular there was no replication of experiments around the same underlying effect. Nevertheless, including replications, as done in previous work (Moonesinghe et al., 2007; https://doi.org/10.1371/journal.pmed.0040028) can make meta-analyses potentially relevant, and we have considered this for future extensions of the model. We now discuss the possible relevance of meta-analyses on page 13 and in our concluding remarks on page 14.

4. In general, I found some explanations of simulation parameters not completely clear. In particular:

a. In the Method section (Model overview subsection) it is unclear how the bias parameter is formalized. Given that it is a crucial element of the simulation approach, especially compared to Ioannidis’s (2005) work. When an experiment is not significant, there is a probability of 1 - b of not publishing. Does this mean that when a non-

significant experiment is repeated with probability b, the significant result (after repetitions) is then considered? The authors should expand the Bias parameter description and implementation within the simulation.

> The reviewer is correct in his interpretation. This was now made clear in the text by adding a sentence on page 17.

b. Why did the authors always fix to 1 the mean of the distributions (except for the “Two Normals” and “Single Normal” models) and why is the minimum effect size of interest set to 0.5? For example, in the social sciences (e.g., Social Psychology) the average effect size is estimated at around 0.3 and a 0.5 as a minimum effect size of

interest seems quite implausible.

> Qualitatively, the absolute values used for the mean and minimum of interest should not make much of a difference in the conclusions, as power is calculated for the average of effects above the minimum. Reducing the minimum will thus lead to higher sample sizes. Results should be similar as long as statistical power is comparable and the shape of the underlying effect size distributions adjusted to keep the prevalence the same. In other words, even if the minimum of interest were to change, there would be combinations of parameters that would provide equivalent results, as many other parameters are defined as functions of the minimum of interest.

We also note that research fields with less subtle interventions and more controlled settings are likely to have greater effect sizes than Social Psychology (see Carneiro et al., 2018, for rodent fear conditioning estimates, for example); thus, it would unfeasible to come up with values that would be realistic for every research area. A consideration about the invariance of results as long as statistical power is calculated based on the minimum effect size of interest has now been added to the methods description on page 16.

5. Related to the previous point, I suggest the inclusion of a simulation flowchart diagram to highlight all relevant steps and parameters. This will improve the manuscript’s readability.

> This was included as Supplementary Figure 1. 

6. In the Sampling section (p.22) is not clear how the author set the sample size based on the desired power level (“Power is specified to detect a typical difference for the field, defined as the average of the effect sizes above the minimum of interest within the model’s distribution of effects (by sampling 100,000 effects from the distribution under study)”). Looking at the source code, the power is analytically calculated using the power.t.test() function but which true effect is used within the function? Could the authors provide a clearer explanation of this step?

> The effect we use for power calculations is indeed the mean of all effects above the minimum of interest. This is considered a “typical” effect size that researchers would be looking for in the field, which would be reflected in the published literature. This is now explained in more detail on page 16.

7. In the Discussion (p.19), the authors point out that simulations’ results are strongly influenced by the assumed effect size distribution (“parameters such as the median effect size inflation in the literature were quite sensitive to the exact shape of the curve, suggesting that estimates from simulations should be taken with caution in the absence of empirical evidence on effect size distributions”). At the same time, it is difficult to estimate the true effect size distribution from published data, especially under some conditions. How do authors suggest dealing with this point? s it possible to quantify the impact of the distribution choice on simulations’ results?

> Large meta-analyses may help approximate effect size curves, but the true underlying distribution will never be known empirically, especially in the presence of publication bias. We advise critically considering different effect size distributions, and interpreting results from a range of plausible parameters to obtain insights. In this sense, we argue that continuous distributions are more plausible than dichotomous ones, which is what led us to undertake this work in the first place. Also, we note that while some parameters such as effect size inflation were more sensitive to distribution, other findings such as the sensitivity of positive predictive values to power, bias and prevalence (i.e. Fig. 2) largely hold Thus, even if simulations are distant from real-world situations and inevitably contain many approximations, they help to gain intuitions on how important we expect these parameters to be for scientific literatures. We have expanded in these points on this passage (page 14).

Minor points

1. I think that Figure 1 could be improved by plotting also the area under the curve according to true/false positive/negative probabilities (for example using transparency for overlapping curves)

> We are not sure we understand the reviewer’s point here. Figure 1 contains the distributions of true effect sizes used in simulations. Those are not estimates for effects taken from the literature, which would contain uncertainty, nor the distribution for experimental samples. Effects thus have a single value and are defined as “true” and true or “false” on the basis of being above or below the minimum. There are thus no true/false positives or negatives at this point: these are only introduced when experiments are simulated (i.e. in the results shown from Fig. 2 onwards).

2. The author used the term “signal” in the context of the effect size direction (type S error). I think that it is more appropriate to use the term “sign” as in the mathematical sense.

> The reviewer is indeed correct on this point. We have corrected changed “signal” to “sign” throughout the text when referring to this type of error. 

3. The simulation’s assumption that results are published even if the effect size is lower than the minimum effect of interest is quite reasonable. I think that an interesting scenario could be to simulate a different publication criterion. Do the authors consider some scenarios where publishing is based on the minimum effect size of interest instead of p-values only?

> We had the same concern as the reviewer, and thus these results had already been included in Figure S2. Overall results are the same, with slight differences occurring only in high power scenarios. With low power, effects below the minimum are generally not significant, so using the minimum as a criterium for publishing does not affect results. This is also mentioned on page 5 of the manuscript.

4. The Github repository contains another simulation parameter called Negative Results Incentive defined as the probability to publish a non-significant result. I would suggest including this parameter in the paper (e.g., in the future directions section) given the current relevance of publishing null findings.

> Exploring the impact of negative results incentive was outside the scope of the current paper, but we included a mention to it in the concluding remarks as a future direction to study on page 14.

Further comments (not required for revision)

I appreciate the effort of organizing and sharing the source code for the Shiny app and the manuscript simulation reproducibility. However, I would suggest improving the repository organization and documentation because I have found it a little bit difficult to follow the simulation steps (especially what is not clear from the manuscript). While the Shiny app is well constructed, the source code is sometimes difficult to understand. Some suggestions:

> We thank the reviewer for the feedback and while we have not made changes to the repository associated with this manuscript, we are still working on the same codebase and their feedback will definitely be reflected in how we organize the repository from now on. 

Reviewer #2: I read and appreciated the article.

Although the results are quite predictable, I think it is important to highlight them.

In this kind of works, it is often necessary to hypersemplify the reality, but I think that the chosen scenarios are quite representative.

I also tried the shiny-app and found it very interesting.

Consequently I don't have particular comments.

Minor points:

- I would have appreciated page numbers in the draft

These have been added, and our responses use that numbering throughout this document.

- page 21 “From this parametrization, we define 5 types of distributions to model

possible scenarios in different scientific fields (see Figure 2 for illustration and Table 2 for

parameters).” It Seems to me that this refers to Figure 1 instead.

> We thank the reviewer for pointing this out. It has now been corrected.

---

## [Decision Letter · Decision Letter 1]

8 Nov 2022

Are most published research findings false in a continuous universe?

PONE-D-22-00486R1

Dear Dr. Neves,

We’re pleased to inform you that your manuscript has been judged scientifically suitable for publication and will be formally accepted for publication once it meets all outstanding technical requirements.

Kind regards,

Antonio Calcagnì, Ph.D.

Academic Editor

PLOS ONE

Additional Editor Comments (optional):

Reviewers' comments:

Reviewer's Responses to Questions

**Comments to the Author**

1. If the authors have adequately addressed your comments raised in a previous round of review and you feel that this manuscript is now acceptable for publication, you may indicate that here to bypass the “Comments to the Author” section, enter your conflict of interest statement in the “Confidential to Editor” section, and submit your "Accept" recommendation.

Reviewer #1: All comments have been addressed

2. Is the manuscript technically sound, and do the data support the conclusions?

Reviewer #1: Yes

3. Has the statistical analysis been performed appropriately and rigorously? 

Reviewer #1: Yes

4. Have the authors made all data underlying the findings in their manuscript fully available?

Reviewer #1: Yes

5. Is the manuscript presented in an intelligible fashion and written in standard English?

Reviewer #1: Yes

6. Review Comments to the Author

Reviewer #1: Authors satisfactorily addressed all my previous comments. I have no further points to discuss.

The only suggestion, but I leave the final decision to authors, is to include the simulation flowchart in the main manuscript instead of in the supplementary materials.

7. PLOS authors have the option to publish the peer review history of their article (what does this mean?). If published, this will include your full peer review and any attached files.

Reviewer #1: No

---

## [Editor Report · Acceptance letter]

9 Dec 2022

PONE-D-22-00486R1 

Are most published research findings false in a continuous universe? 

Dear Dr. Neves:

I'm pleased to inform you that your manuscript has been deemed suitable for publication in PLOS ONE. Congratulations! Your manuscript is now with our production department. 

Kind regards, 

on behalf of

Dr. Antonio Calcagnì 

Academic Editor

PLOS ONE